# Effect of a New Formulation of Nutraceuticals as an Add-On to Metformin Monotherapy for Patients with Type 2 Diabetes and Suboptimal Glycemic Control: A Randomized Controlled Trial

**DOI:** 10.3390/nu13072373

**Published:** 2021-07-11

**Authors:** Giovanni Sartore, Eugenio Ragazzi, Giulia Antonello, Chiara Cosma, Annunziata Lapolla

**Affiliations:** 1Department of Medicine-DIMED, University of Padova, I-35128 Padova, Italy; g.sartore@unipd.it (G.S.); antonello.giulia@gmail.com (G.A.); annunziata.lapolla@unipd.it (A.L.); 2Department of Pharmaceutical and Pharmacological Sciences, University of Padova, I-35131 Padova, Italy; 3Department of Laboratory Medicine, University of Padova, I-35128 Padova, Italy; chiara.cosma@sanita.padova.it

**Keywords:** advanced glycation end products, berberine, hesperidin, glyco-oxidation

## Abstract

The aim of the study was to evaluate the overall biohumoral and metabolic effects of a 12-week add-on therapy consisting of a new nutraceutical formulation (BHC) based on berberine, hesperidin, and chromium picolinate in type 2 diabetes mellitus (T2D) patients with suboptimal glycemic compensation receiving metformin. After 12 weeks, participants in the group receiving metformin plus BHC, compared to the group receiving metformin only, saw a significant improvement in their glucose profile, in terms of both glycated hemoglobin (HbA1c) and fasting blood glucose (FBG). Their FBG dropped from 145 ± 20 mg/dL to 128 ± 23 mg/dL (*p* < 0.01), a decrease of 11.7% compared with the baseline. This decrease differed significantly from the situation in the control arm (*p* < 0.05). HbA1c decreased by 7.5% from the baseline, from 53.5 ± 4.3 mmol/mol to 49.5 ± 5.1 mmol/mol (*p* < 0.01), in the group given BHC, while no difference was seen in the control group. Advanced glycation end products (AGEs) and malondialdehyde (MDA) were found to be significantly reduced (*p* < 0.01) only in the BHC group, from 9.34 ± 7.61 μg/mL to 6.75 ± 6.13 μg/mL, and from 1.7 ± 0.15 μmol/L to 1.4 ± 0.25 μmol/L, respectively. In patients with T2D taking metformin with suboptimal glycemic compensation, adding BHC for 3 months significantly improved glucose control in terms of FBG and HbA1c, and had a positive effect on the lipid peroxidation profile, as indicated by a decrease in AGEs and MDA.

## 1. Introduction

The pathophysiology of type 2 diabetes mellitus (T2D) involves increased peripheral insulin resistance and a loss of pancreatic β-cell function (resulting in reduced insulin secretion) as the main contributors to a gradually deteriorating glycemic control. An excess of oxidative stress (in terms of glyco-oxidation and lipid peroxidation) and low-grade systemic inflammation are recognized as the main determinants of β-cell dysfunction [1,2]. Patients with T2D also feature increased oxidative stress in terms of advanced glycation end products (AGEs) and malondialdehyde (MDA), which contribute—together with chronic hyperglycemia—to the pathophysiology of atherosclerosis [2].

Preserving β-cell function and reducing oxidative stress are, therefore, essential to the management of T2D. Metformin is considered the treatment of choice at the onset of T2D. The guidelines and consensus recommend its use for the treatment of hyperglycemia. In approximately 30% of patients initially responsive to metformin, this treatment no longer suffices after 3 years, and these patients must be given a second hypoglycemic agent [3]. This is an effect of the progression of the disease over time.

Experimental and clinical studies have shown that nutraceutical supplements are effective in inhibiting intracellular inflammatory cascades, the formation of oxygen free radicals, and the production of pro-oxidant substances in T2D [1].

In particular, berberine (a benzylisoquinoline alkaloid obtained from *Berberis* species) has shown several effects: it modulates inflammation and oxidative stress in both animal models and humans; it has insulin-sensitizing effects in patients with T2D [4,5]. Berberine also exerts a documented hypoglycemic and hypocholesterolemic effect, as demonstrated by in vitro and in vivo studies. It is one of the molecules with the best-documented hypoglycemic activity, and it improves glucose tolerance. Its pharmacological and biochemical mechanisms of action have been established [6,7].

Several scientific works have highlighted various mechanisms underlying the hypoglycemic effect of berberine in T2D. It has been shown to promote insulin secretion and improve β-cell function in the pancreas, to increase the secretion of glucagon-like protein-1 in the intestine, and to reduce the intestinal absorption of glucose, to regulate glycolipid metabolism and reduce gluconeogenesis in the liver by modulating the expression of peroxisome proliferator-activated receptors, and to stimulate glycolysis in peripheral cells [8,9,10,11]. 

Berberine also possesses antioxidant properties and appears to be involved in the mechanisms that modulate the intestinal flora [12,13,14]. It is poorly absorbed by the intestine and has a limited oral bioavailability [15] due to the presence of permeability glycoproteins (P-gp)—membrane glycoproteins with a pump function—that act as a functional barrier by expelling berberine from the cell towards the intestinal lumen. To overcome this problem and improve the bioavailability of berberine, the tablet is formulated with Enterosoma^TM^ Technology (Labomar patent), a patented chitosan salt that interacts with P-gp, limiting the expulsion of berberine from the cell, and favoring its entry into the circulation [16].

Hesperidin is a glycosylated flavonoid widely occurring in citrus fruits. It accounts for 90% of all orange flavonoids. It has vasoprotective properties and appears to play a key part in countering micro-inflammation and oxidative stress at the cell level. The mechanism of action of hesperidin and its metabolite, aglycone hesperetin, has been investigated in numerous studies. Citrus fruit consumption is certainly associated with a reduced risk of cardiovascular diseases [17,18], and less oxidative damage to DNA in plasma cells [19,20]. Recent clinical studies have shown a correlation between the intake of hesperidin and endothelial function (with an increase in endothelium-dependent vascular reactivity, and changes in the concentration of endothelial activation markers) and circulating inflammation markers [21]. Hesperidin has revealed interesting pharmacological antitumor, lipid-lowering, and antioxidant properties in animal models of T2D [22]. A hypoglycemic and lipid-lowering effect of this nutraceutical has been envisaged too, again in diabetic animal models [23], through the modulation of important enzymes such as PPAR and HMG-CoA reductase.

According to some studies, chromium appears to be able to enhance the action of insulin, improving glucose tolerance as a result [24,25,26,27]. The usefulness of chromium in treating patients with T2D, and its mechanism of action are still under study, however. Great attention recently focused on chromium supplementation in animal models and humans with T2D, and its effects on glycemic control parameters and markers of inflammation and oxidative stress [28].

No clinical studies are available on the hypoglycemic or antioxidant activity of hesperidin in patients with T2D. The anti-inflammatory and antioxidant effects—and possible improvement in glycemic and lipid compensation—of berberine/hesperidin in association with metformin could slow the process of β-cell deterioration (delaying the failure of primary and secondary oral hypoglycemic agents), and counteract the atherosclerotic processes that are often already present. Supplementation with chromium picolinate is also important in a nutraceutical compound intended for patients with T2D, given its insulin-sensitizing action and adjuvant effect on glycemic control.

The aim of the study was to evaluate the overall metabolic effects of a 12-week add-on therapy with a new nutraceutical formulation (BHC) based on berberine, hesperidin, and chromium picolinate in T2D patients with suboptimal glycemic compensation receiving metformin. The primary endpoint considered changes in glycated hemoglobin (HbA1c) levels and in fasting blood glucose (FBG) compared with baseline values. Secondary goals concerned the possible effects of BHC in improving clinical, biohumoral, and metabolic parameters (lipid profile, BMI, insulin resistance indices, and β-cell function), and in reducing lipid peroxidation (MDA), glyco-oxidation (AGEs, sRAGE), and inflammation markers (hsCRP, IL-1, IL-6, TNFα).

## 2. Materials and Methods

### 2.1. Participants and Study Design

This was a single-center, randomized (1:1), open, controlled study. Participants were recruited at the Diabetes Clinic of the ULSS 6 Padova (north-east Italy). They were enrolled on the basis of the two latest HbA1c levels obtained in tests performed up to 6 months prior to their baseline examination. Both groups adopted the same standard diet for diabetes, adjusted for individual energy needs, for the duration of the study. Patients were instructed to follow the usual physical activity for all the investigation period. Comparison of data by means of a paired statistical test allowed to take account of any subjective variations among patients, allowing to evaluate the net effect of the treatment. On the basis of a randomization procedure, participants were assigned to the group assuming metformin alone or in the group receiving metformin plus the product under study. The add-on product was to be taken for 12 weeks, and participants were given precise indications on how to take it, its dosage, and storage. 

Individuals who met the following inclusion criteria at the baseline visit were considered eligible for the study: Caucasians, of both sexes, aged ≥18 and ≤65 years; who provided their written informed consent; on metformin therapy; achieving suboptimal glycemic control, judging from two HbA1c tests indicating between 6.5% and 7.5% in the previous 6 months; able to understand the nature, purposes, and procedures of the study.

Subjects meeting the following criteria were excluded from the study: type 1 diabetes (or LADA); gestational diabetes; severe micro- or macrovascular complications of T2D (proliferative retinopathy, chronic renal insufficiency, ischemic heart disease, lower limb arterial disease, ischemic stroke, neuro/ischemic ulcers); active smokers; chronic inflammatory or autoimmune diseases; pregnancy or lactation (women of childbearing age had to produce a negative pregnancy test result before they were administered the product under study); previous major abdominal surgery or other pathological conditions that might impair gastrointestinal absorption; eating disorders; treatment with statins and/or fibrates for less than 6 months; alcohol or drug abuse; use of food supplements, particularly (but not only) those containing fibers and polysaccharides, in the previous 6 months, with a frequency and dosage sufficient to interfere with the study.

### 2.2. Nutraceutical Formulation 

The product used for the study was supplied by Labomar srl (Istrana, Italy) in the form of coated tablets formulated with Enterosoma^TM^ Technology (chitosan salt; Labomar patent). The active ingredients are *Berberis aristata* DC. extract 250 mg, titrated at 97% in berberine hydrochloride (HCl), hesperidin 200 mg (from *Citrus aurantium* L. fruit), and chromium picolinate (200 mcg). Enterosoma^TM^ consists of a gastro-resistant tablet containing a chitosan polymer and N-acetylcysteine in the inner core, together with the active ingredients. This enables enterocyte-associated Pg-P activity modulation and tight junction cohesion lowering [16], thus enhancing the bioavailability of the active ingredients. Each subject received 1 tablet SID for the 12-week study period.

### 2.3. Measurements

FBG, total cholesterol, HDL, LDL, and triglycerides were measured with reagents of Roche Diagnostics (reference catalog 05168791, 05168538, 07528582, 07005768, and 051714017, respectively) in a fully automatic platform Cobas C 702 (Roche Diagnostics).

HbA1c was measured by high-performance liquid chromatography (HPLC, Menarini Akray ADAM A1c HA-8180v), in line with IFCC standards (International Federation of Clinical Chemistry), whereas MDA (reference catalog 67000) was measured using a Chromsystems kit.

IL-1β, TNFα, and IL-6 (reference catalog LKL11, LKNF1, and LK6P1, respectively) were measured using platform Immulite One (Siemens), whereas Insulin, and C-peptide (reference catalog L2KIN2, and LKNF1, respectively) using the Immulite 2000 analyzer (Siemens). hsCRP (reference catalog K7046) was measured using the fully automatic instrument Dimension Vista 1500 (Siemens).

AGE and sRAGE were measured using an ELISA kit (reference catalog CSB-E09412h, Cusabio and reference catalog RD191116200r, BioVendor, respectively). 

### 2.4. Statistical Analysis

Continuous variables are reported as means ± standard deviations (SD). The effectiveness of the treatment was assessed with the Student’s *t*-test for independent data (comparisons between the two treatments), or for paired data (comparisons within each treatment), considering the variations between time 0 and at 12 weeks. Fisher’s exact test was used for categorical data. Significance was accepted for *p* < 0.05. The analysis was conducted using JMP Pro software, version 14 (SAS Institute Inc., Cary, NC, USA).

## 3. Results

Judging from our patients’ reports, the BHC was readily accepted and well tolerated as an add-on to their metformin treatment. No adverse events were recorded.

Table 1 shows the baseline clinical characteristics of the patients in the intervention and control groups.

After 12 weeks, the two groups’ anthropometric parameters remained unchanged (Table 2). A reduction in fasting glucose (*p* < 0.01) was observed only in the group given BHC in addition to metformin, and the difference was significant compared with subjects in the control group treated with metformin alone (*p* < 0.05). Glycated hemoglobin was also significantly reduced in the group taking BHC (*p* < 0.01, compared with the baseline; *p* < 0.01 compared with the control group).

There were no significant changes in either group as regards lipid profile, c-peptide, fasting insulinemia, or HOMA-IR.

At the end of the 12-week period, only the BHC group showed a significant decrease in MDA and AGEs (*p* < 0.01), with a significant difference vis-à-vis the control group (*p* < 0.005 and *p* < 0.05, respectively). On the other hand, s-RAGEs remained unchanged at the end of the study in both groups compared with the baseline.

As for the inflammatory indicators explored (TNFα, IL-1, IL-6, and hsCRP), there were no significant intra- or intergroup differences between before and after the treatment.

## 4. Discussion

Adding BHC—a new nutraceutical formulation with an improved bioavailability—in the treatment of T2D patients with a suboptimal glycemic control on metformin monotherapy resulted in lower HbA1c and FBG levels, according to the primary endpoint. This reduction was less conspicuous than in other reports on berberine, probably because we used 250 mg of berberine a day, which is just a quarter of the dosage commonly adopted in other studies [29]. It is important to emphasize that even such a low dosage of berberine was able to influence patients’ glycemic control. This result is even more interesting in patients such as ours, whose baseline HbA1c levels were already low. Our results support also experimental in vivo investigations [30] that demonstrated the antidiabetic activity of berberine.

Berberine can also affect the lipid profile, as suggested by clinical reviews [31]. When used at doses of 1–1.5 g/day, it prompted a significant reduction in total cholesterol, LDL-cholesterol, and triglycerides [31]. The lack of this effect on the lipid profile in our study is probably due to the low dose of berberine used, and the fact that nearly two in three of our participants were taking statins.

The BHC formulation used in the present study also contains chromium picolinate as it has been suggested that chromium supplementation may have a modest beneficial effect on glycemia in patients with T2D [25]. This trace element may have contributed to our findings because some (but not all) authors found it effective at doses of 200 μg or more [25]. As BHC consists of a combination of nutraceuticals, there is no way to distinguish between the effects of the single components. It has been hypothesized, however, that chromium can only have an effect in reducing HbA1c and FBG in patients with a particular T2D phenotype, i.e., HbA1c > 64 mmol/mol (8%), FBG > 9.7 mmol/L (175 mg/dL), a marked insulin resistance, and a short history of diabetes (less than 5 years or so) [32,33].

The reason for including hesperidin in the nutraceutical product tested here is to take action on a patient’s lipids, glyco-oxidation, and inflammation. Our results regarding MDA levels are highly significant, and very intriguing because they indicate a decrease in lipid peroxidation, in line with the result reported by Homayouni et al. [34]. 

Many natural AGE inhibitors are available as dietary supplements. Hesperidin is a polyphenol that has an anti-glycation activity through various mechanisms, including antioxidant effects and the inhibition of aldose reductase, an enzyme in the polyol pathway that is enhanced under hyperglycemic conditions [35]. The polyol pathway is an important source of diabetes-induced oxidative stress, and contributes to AGE accumulation [36]. These mechanisms could lie behind the positive results obtained in our study as regards AGEs. That said, the supplement given to our intervention group did not affect their circulating sRAGE levels, which are more influenced by changes in the inflammatory state (that were not seen in our patients). It is worth noting, however, that our control group had significantly higher TNFα levels than our BHC-treated group at the end of the 12 weeks.

As regards inflammatory parameters, Rizza et al. [21] recorded a significant reduction in hsCRP when they gave patients 500 mg of hesperidin, while Morand et al. [20] saw no significant decrease with a dosage of about 300 mg. Our findings appear to be in line with the latter.

The strength of the present investigation lies in that it underscores the possible implications of modulating lipid peroxidation and glyco-oxidation with nutraceuticals containing berberine and hesperidin in patients with T2D, administered as an add-on to a first-line therapy such as metformin. The results provide a clinical validation of the role of the compounds in mediating beneficial effects on cardiovascular diseases, as suggested by in vivo experiments on animal models [37]. Moreover, the fact that a low dose of both berberine and hesperidin can offer a favorable outcome in T2D patients suggests an additional strategy to manage the complications of the disease, with no appreciable side effects.

The main limitations of our study concern the small number of patients considered, and the low doses of the compounds investigated—though we consider it important to explore the efficacy of nutraceutical products at lower doses than those already found effective. Our data suggest that BHC supplementation for 3 months has a positive effect on glyco-oxidation and lipid peroxidation in patients with T2D.

It would be interesting to extend the study to a larger sample and for a longer period of time to assess the long-term effects of reducing lipid peroxidation and glyco-oxidation in patients with T2D.

## Figures and Tables

**Table 1 nutrients-13-02373-t001:** Baseline clinical characteristics of the intervention and control groups. For continuous data, values are presented as mean ± SD.

Parameter	Add-On Group (*n* = 20)	Control Group (*n* = 20)	*p* ^†^
Age, years	67.2 ± 6.3	66.9 ± 6.1	ns
Gender, M/F	15/5	17/3	ns
Duration of diabetes, years	8.3 ± 3.5	7.9 ± 6.1	ns
Smoker, Yes/No	3/17	4/15	ns
Metformin dose, mg/day	1295 ± 538	1450 ± 777	ns
Antihypertensive therapy, Yes/No	14/6	15/5	ns
Statins, Yes/No	13/7	13/7	ns

^†^ Statistical analysis was performed with the Student’s *t*-test for continuous variables, and Fisher’s exact test for categorical data. ns: not statistically significant.

**Table 2 nutrients-13-02373-t002:** Hematochemical parameters of patients at the baseline and after 12 weeks of treatment. For continuous data, values are mean ± SD.

Parameter	Add-On Group (*n* = 20)		Control Group (*n* = 20)		*p*^†^ (Between Groups)	
	Before	After	Before	After	Before	After
IFCC-HbA1c, mmol/mol (DCCT-HbA1c, %)	53.5 ± 3.7 (7.0 ± 2.5)	49.5 ± 5.1 * (6.7 ± 2.6)	53.9 ± 4.3 (7.1 ± 2.5)	56.4 ± 4.3 (7.3 ± 2.5)	ns	<0.01
FBG, mg/dL	145 ± 20	128 ± 23 *	150 ± 32	152 ± 35	ns	<0.05
AGEs, μg/mL	9.34 ± 7.61	6.75 ± 6.13 *	9.02 ± 5.37	12.79 ± 7.71	ns	<0.05
s-RAGEs, pg/mL	597 ± 188	815 ± 805	566 ± 139	535 ± 145	ns	ns
Total cholesterol, mg/dL	166 ± 41	167 ± 32	158 ± 29	159 ± 38	ns	ns
HDL cholesterol, mg/dL	50 ± 12	48 ± 11	52 ± 18	52 ± 18	ns	ns
LDL cholesterol, mg/dL	92 ± 37	83 ± 40	80 ± 23	77 ± 33	ns	ns
Triglycerides, mg/dL	123 ± 63	136 ± 79	133 ± 108	129 ± 112	ns	ns
Fasting C-peptide, nmol/L	3.3 ± 1.2	3.4 ± 1.5	3.6 ± 2.0	3.5 ± 1.7	ns	ns
Fasting serum Insulin, pmol/L	17.0 ± 15.1	27.6 ± 43.6	16.7 ± 14.7	15.0 ± 9.3	ns	ns
HOMA -IR	5.58 ± 4.62	8.27 ± 12.19	6.91 ± 10.05	5.03 ± 3.52	ns	ns
MDA, μmol/L	1.7 ± 0.15	1.4 ± 0.25*	1.7 ± 0.21	1.7 ± 0.29	ns	<0.005
IL-1, pg/mL	1.78 ± 0.64	2.09 ± 0.83	1.78 ± 0.62	2.16 ± 0.45	ns	ns
IL-6, pg/mL	3.5 ± 2.1	4.9 ± 4.6	2.8 ± 0.6	2.9 ± 0.9	ns	ns
TNFα, pg/mL	8.6 ± 5.1	7.8 ± 2.9	6.8 ± 1.6	7.2 ± 1.6	ns	ns
hsCRP, mg/L	1.46 ± 1.25	1.39 ± 2.33	1.18 ± 0.78	1.64 ± 1.26	ns	ns
Systolic blood pressure, mmHg	136 ± 12	136 ± 12	138 ± 10	137 ± 9	ns	ns
Diastolic blood pressure, mmHg	82 ± 8	82 ± 8	81 ± 7	81 ± 11	ns	ns
BMI, kg/m^2^	29.1 ± 5.2	29.5 ± 5.3	28.7 ± 4.2	28.8 ± 4.2	ns	ns
Waist circumference, cm	102 ± 9	102 ± 9	101 ± 10	101 ± 9	ns	ns

* *p* < 0.01. Statistical analysis comparing values before vs after intervention was performed with the Student’s *t*-test for paired data; in the case of not normally distributed data, the non-parametric test of Wilcoxon was used. ^†^ Comparison between groups was performed with the Student’s *t*-test for independent (unpaired) data. ns: not statistically significant. AGEs: advanced glycation end products; BMI: body mass index; DCCT-HbA1c: Diabetes Control and Complications Trial units for glycated hemoglobin; FBG: fasting blood glucose; HDL: high-density lipoprotein; HOMA-IR: Homeostatic Model Assessment for Insulin Resistance; hsCRP: high sensitivity C-reactive protein; IFCC -HbA1c: International Federation of Clinical Chemistry units for glycated hemoglobin; IL-1: Interleukin-1; IL-6: Interleukin-6; LDL: low-density lipoprotein; MDA: malondialdehyde; s-RAGE: soluble receptor of advanced glycation end products; TNFα: Tumor necrosis factor alpha.

## Data Availability

Not applicable.

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
