# Peer review of "Effect of a New Formulation of Nutraceuticals as an Add-On to Metformin Monotherapy for Patients with Type 2 Diabetes and Suboptimal Glycemic Control: A Randomized Controlled Trial"

_nutrients, 2021, doi:10.3390/nu13072373_

Round 1
Reviewer 1 Report
The authors describe their work on the effects of a new formulation of berberine, hesperidin and chromium (BHC) as an adjunct to metformin monotherapy in type 2 diabetics. It was found that the BHC formulation improved the atherogenic profile in terms of FBG, A1c AGEs and MDA. This is an interesting, but preliminary study. Appropriate methodology has been employed and the conclusions appear to be justified based on the data at hand. However, I have a few recommendations for consideration.
- Abstract. Last sentence. I would refrain from the use of atherogenic profile and refer more to glucose handling and oxidative stress.
- Introduction. Need a stronger rationale for the study as well as a clear hypothesis to be tested in the study.
- Results. Table 2. Some of the data that is indicated as significantly different does not appear to be the case, please check statistical validation of the data.
- Results. Since there are some females in the study, is it possible to reanalyze data according to sex even though small sample size, but there may be some trend in the response to BHC based on sex.
- Results. The data for the control group is confusing, at least to this reviewer, no effect of metformin on FPG? Also, AGEs and A1c in this group are increased Any comment?
- Discussion. Effects of other medications and co-morbidities should be described.
- Discussion. The authors need to elaborate and emphasize the novelty aspect of their work as well as on the potential clinical applicability of the findings.
Reviewer 2 Report
I read the article written by Sartone G, et al.
I understand BHC supplement was effective on improving glucose homeostasis in diabetes patients.
I ask whether the authors investigated food calorie intake and locomotor activity, resting energy expenditure on the patients. The authors should get and add those data in before and after the administration.
I wonder why the insulin resistance index was worsen in the patients treated with BHC. I would like to know that reason.
How about adverse effects of BHC. Did the author measure serum liver enzymes, amylase, urinalysis and monitor the EKG?
Reviewer 3 Report
The authors described the hypoglycemic effect of a new nutraceutical formulation based on berberine as an add-on to metformin for patients with T2D. The manuscript is well written, with only some minor tidy-up.
Minor points:
- In the Abstract, the first sentence is too long and difficult to read and Conclusions should be clearly identified. In addition, atherogenic profile is more than FBG, HbA1c, AGEs and MDA. Therefore, this conclusion goes beyond the authors’ results.
- HbA1c levels in percentage have been used for the inclusion of patients in the study. Thus, changes showed in table 2 should be also reported in percentage.
- The dosage of the nutraceutical a day should be clearly reported in Methods section.
Round 2
Reviewer 1 Report
The authors have addressed all initial concerns adequately. I have no further comments.
Reviewer 3 Report
.